# Genome Sequence of *Eubacterium limosum* B2 and Evolution for Growth on a Mineral Medium with Methanol and CO_2_ as Sole Carbon Sources

**DOI:** 10.3390/microorganisms10091790

**Published:** 2022-09-05

**Authors:** Guillaume Pregnon, Nigel P. Minton, Philippe Soucaille

**Affiliations:** 1INSA, UPS, INP, Toulouse Biotechnology Institute (TBI), Université de Toulouse, 31400 Toulouse, France; 2BBSRC/EPSRC Synthetic Biology Research Centre (SBRC), School of Life Sciences, University Park, The University of Nottingham, Nottingham NG7 2RD, UK

**Keywords:** *Eubacterium limosum*, acetogen, adaptive laboratory evolution, genome sequence, methanol, butyric acid, mineral medium, proteomics

## Abstract

*Eubacterium limosum* is an acetogen that can produce butyrate along with acetate as the main fermentation end-product from methanol, a promising C1 feedstock. Although physiological characterization of *E. limosum* B2 during methylotrophy was previously performed, the strain was cultured in a semi-defined medium, limiting the scope for further metabolic insights. Here, we sequenced the complete genome of the native strain and performed adaptive laboratory evolution to sustain growth on methanol mineral medium. The evolved population significantly improved its maximal growth rate by 3.45-fold. Furthermore, three clones from the evolved population were isolated on methanol mineral medium without cysteine by the addition of sodium thiosulfate. To identify mutations related to growth improvement, the whole genomes of wild-type *E. limosum* B2, the 10th, 25th, 50th, and 75th generations, and the three clones were sequenced. We explored the total proteomes of the native and the best evolved clone (n°2) and noticed significant differences in proteins involved in gluconeogenesis, anaplerotic reactions, and sulphate metabolism. Furthermore, a homologous recombination was found in subunit S of the type I restriction-modification system between both strains, changing the structure of the subunit, its sequence recognition and the methylome of the evolved clone. Taken together, the genomic, proteomic and methylomic data suggest a possible epigenetic mechanism of metabolic regulation.

## 1. Introduction

Strictly anaerobic acetogenic bacteria are known to fix one carbon (C1) substrates, such as synthesis gas (syngas) CO_2_ + H_2_ or CO, using the methyl and carbonyl branches of the Wood–Ljungdahl pathway (WLP). Syngas metabolization consists of the reduction of two molecules of CO_2_ with H_2_ as an electron donor or by incorporation of CO in the carbonyl branch by the carbon monoxide dehydrogenase/acetyl-CoA enzyme and results in the production of one molecule of acetyl-CoA [1]. This central intermediate is mainly used to produce ATP and acetate. However, depending on the acetogen, various molecules can be produced from acetyl-CoA, such as ethanol, or more complex products, including lactate, butyrate, butanol or 2,3-butanediol [2,3]. Conversion of C1 compounds to four-carbon products, such as butyrate, by microorganisms has received much attention due to its potential application in numerous fields, including the energy, cosmetics, and pharmaceutical industries [4]. Moreover, butyryl-CoA, an intermediate in the butyrate pathway, can be reduced to n-butanol, an attractive molecule that can be used both as a platform chemical and a biofuel [5]. Driven by growing ecological interest, syngas exploitation by bacteria is booming with the development of the first factories of industrial scale [6]. However, gas substrate utilization faces several technical challenges, such as storage and gas–liquid mass transfer, decreasing bacterial productivity [7]. Methanol constitutes an attractive C1 carbon source due to its wide availability throughout the world, easy production process, storability, and liquid state to avoid mass transfer issues [8,9]. The acetogen *Eubacterium limosum* can metabolize methanol and remains one of the few bacterial species able to convert it to butyrate with the coproduction of acetate [10]. Although numerous physiological studies of *E. limosum* B2 have been performed in the past [10,11,12,13,14], its genome sequence is still undetermined, and the first biochemical characterization of anaerobic methanol metabolism, including the energetic aspect, was performed in the model methylotrophic bacterium *Acetobacterium woodii* [15], which does not produce C4 compounds. Two molecules of methanol are converted to two molecules of methyl-THF by a methyltransferase system. One molecule is oxidized into CO_2_ through the reverse use of the methyl branch to produce the reducing equivalents needed to then reduce CO_2_ to CO, while the second molecule is used with CO by the ACS/CODH complex to produce acetyl-CoA. This method of methanol assimilation through the WLP was also demonstrated for *Eubacterium callanderi* KIST 612, formerly named *E. limosum* KIST612 [16]. Moreover, the first evidence of butanol production by *E. limosum* ATCC 8486 from methanol and formate used as cosubstrates was recently obtained, reinforcing the interest in this bacterial species [17]. Methanol metabolism by *E. limosum* B2 was characterized in semi-defined medium containing yeast extract (YE) and cysteine [10]. To the best of our knowledge, no study has been performed on any *E. limosum* strain to develop a mineral medium for growth on methanol as a carbon and energy source. A strain that can grow on a mineral medium will have several advantages for in-depth metabolic characterization using flux balance analysis and genome-scale models [18]. The semi-defined medium previously used for *E. limosum* B2 contained YE but also cysteine, as the strain was shown to be an auxotroph for cysteine [10,19]. Synthetic medium free from YE is also of economic interest, as YE is an expansive supplement prohibiting favourable economic scale-up [20]. Growth improvement can be easily achieved by adaptive laboratory evolution (ALE), which consists of culturing a microorganism for several generations in restrictive medium to naturally improve its growth parameters [21]. Adaptive laboratory evolution has already been performed on *E. limosum* ATCC 8486 using CO as a carbon and energy source, improving its growth rate 1.4-fold [22]. Here, we aimed to deepen our knowledge of the *E. limosum* B2 strain at the genome level and to develop a mineral medium allowing good growth on methanol of an evolved strain for further performing systems biology analysis in chemostat culture. For this purpose, we (i) performed de novo whole-genome sequencing by single-molecule real-time technology (SMRT), (ii) developed a defined medium allowing the strain to grow with methanol as a carbon and energy source without the addition of YE but with cysteine, (iii) improved the growth performance of the strain on this medium by ALE, (iv) developed a mineral medium allowing the adapted strain to grow with methanol as a carbon and energy source without the addition of cysteine, (v) studied genomic modifications that occurred during the evolution process and (vi) performed a comparative study of the total proteome between an isolated clone from the evolved population and the wild-type strain.

## 2. Materials and Methods

### 2.1. Bacterial Strain and Growth Conditions

The *E. limosum* B2 strain used in this study has been studied in our laboratory since 1985 and is stored at −80 °C. Culture growth was performed in serum bottles (in 25 mL of culture medium) under a strict anaerobic nitrogen (N_2_) atmosphere and incubated at 37 °C at pH 7.4. The wild-type strain was cultured in rich medium for anaerobic bacteria (M187), which comprised the following components: tryptone 30 g/L, yeast extract 20 g/L, glucose 5 g/L, and cysteine hydrochloride 0.5 g/L. The adapted strain in the methanol substrate was cultured in synthetic medium composed of the following components: NaCl 0.6 g/L, NH_4_Cl 1 g/L, MgSO_4_-7H_2_O 0.12 g/L, CaCl_2_ 0.08 g/L, KH_2_PO_4_ 1 g/L, vitamin solution 100X (biotin 20 mg/L, pantothenic acid 50 mg/L, lipoic acid 50 mg/L), sodium thiosulfate pentahydrate 4 mM, potassium carbonate 1 g/L, titanium citrate 2 mM as a reducing agent, and trace element solution 100X (MnCl_2_ 0.1 g/L, CoCl_2_-2H_2_O 0.1 g/L, ZnCl_2_ 0.1 g/L, CuCl_2_ 0.02 g/L, H_3_BO_3_ 0.01 g/L, Na molybdate 0.01 g/L, Na_2_SeO_3_ 0.017 g/L, NiSO_4_-6H_2_O 0.026 g/L, FeSO_4_,6H_2_O 1 g/L and nitrilotriacetic acid 12.8 g/L). Resazurin was added at 1.10^−^^4^% (*wt*/*wt*) as a redox potential indicator. Glucose (6 g/L) and methanol (6.4 g/L) were the two substrates tested, corresponding to 200 mM carbon.

Isolated clones from the methanol-adapted population were cultured in solid synthetic medium with glucose as the substrate but without cysteine. The product concentrations for solid synthetic medium elaboration remained the same as those for liquid synthetic medium supplemented with 15 g/L of agarose. After solidification, plates were placed in an anaerobic chamber overnight before streaking the population to ensure balance of the anaerobic atmosphere on the solid medium (90% N_2_, 5% CO_2_ and 5% H_2_).

### 2.2. Adaptive Laboratory Evolution

The ALE was performed from the first generation of a yeast extract-free adapted *E. limosum* B2 strain previously obtained in the laboratory in synthetic medium with 200 mM methanol as the substrate (unpublished data). A 5% volume of culture was transferred to fresh medium every three generations. The population was cultured over 100 generations.

### 2.3. Clone Isolation on Methanol Mineral Medium without Cysteine

Single clones from the adapted population (over 100 generations) were isolated in solid glucose mineral medium without cysteine in an anaerobic chamber at 37 °C. Cysteine was replaced by 4 mM sodium thiosulfate in the synthetic medium. Three clones were cultivated in liquid methanol mineral medium without cysteine to follow and compare the growth profile from the 10th, 25th, 50th and 75th generations adapted in MMM with cysteine. Cysteine was replaced by 4 mM sodium thiosulfate.

### 2.4. Analytical Methods

Culture density was measured by spectrophotometry at 620 nm (OD620) against a water blank. Growth rates were calculated from these measurements. Substrates and products were measured using a high-performance liquid chromatography instrument (Agilent 1200 series, Massy, France) equipped with a refractive index detector and Aminex HPX 87 H 300 × 7.8 mm. The acid H_2_SO_4_ (0.5 mM) was used as the mobile phase and elution was performed at 48 °C. The biomass formula C_4_H_7_O_2_N_0.6_ was used to convert biomass values into molar cell carbon concentration (mC) [23]. Acetate and butyrate production yields were calculated in millimolar carbon from the product per 100 mM carbon from methanol (mC/100 mC MetOH).

### 2.5. Genomic DNA Extraction

For de novo sequencing, the wild-type strain of *E. limosum* B2 was cultured in rich medium for anaerobic bacteria (M187). Genomic DNA of harvested cells in mid-exponential phase from 1 mL culture was carried out with the Wizard^®^ HMW DNA extraction kit (Promega, Madison, WI, USA) according to the manufacturer’s instructions with the following modifications. The lysis step with lysozyme treatment was extended to 2 h to optimize the DNA extraction yield and genomic DNA was resuspended in 10 mM Tris-HCl (pH 8.5) solution. The DNA quantification was performed spectrophotometrically by a Nanodrop 2000 spectrophotometer (Thermo Fisher Scientific, Wilmington, DE, USA) and fluorometrically using a Qubit^TM^ instrument (Thermo Fisher Scientific, Wilmington, DE, USA). Quality assessment was performed by (i) gel electrophoresis, (ii) the A260/A280 and A260/230 ratios (>1.9) and (iii) DNA concentration comparison by Qubit and Nanodrop. The difference between the results from both instruments had to be less than 2 for validation.

For genomic comparison of the evolved population through generations, whole-genomic DNA of the wild-type strain, the 10th, 25th, 50th, and 75th generations and three isolated clones was extracted using the same kit as that used for the WT strain for de novo sequencing, measured by Nanodrop and stored at −20 °C until sequencing.

### 2.6. Whole Genome Sequencing

Single-molecule real-time long read sequencing of the *E. limosum* B2 wild-type strain was performed at the Gentyane Sequencing Platform (Clermont-Ferrand, France, https://gentyane.clermont.inrae.fr, accessed on 25 November 2020) with a PacBio Sequel II Sequencer (Pacific Biosciences, Menlo Park, CA, USA). The SMRTBell library was prepared using a SMRTbell Express 2 Template prep kit (Pacific Biosciences, Menlo Park, CA, USA), following the “procedure and checklist -preparing Multiplexed Microbial Libraries using SMRTbell Express Template prep kit 2.0” protocol. Genomic DNA (1 µg) of each strain was sheared using g-tubes (Covaris, England), generating DNA fragments of approximately 10 kb. A Fragment Analyzer (Agilent Technologies, Santa Clara, CA, USA) assay was used to assess the fragment size distribution. Sheared genomic DNA was carried into the enzymatic reactions to remove the single-strand overhangs and to repair any damage that may be present on the DNA backbone. An A-tailing reaction followed by overhang adapter ligation was conducted to generate SMRTbell templates. The sample was then purified with 0.45X AMPure PB beads to obtain the final libraries of approximately 10 kb. The SMRTBell libraries were inspected for quality and quantified on a Fragment Analyzer (Agilent Technologies, Santa Clara, CA, USA) and with a Qubit fluorimeter with the Qubit dsDNA HS reagent Assay kit (Life Technologies, Eugene, OR, USA). A ready-to-sequence SMRTBell Polymerase Complex was created using a Binding Kit 2.0 (Pacific Biosciences, Menlo Park, CA, USA) and the primer V4, and the diffusion loading protocol was used according to the manufacturer’s instructions. The PacBio Sequel instrument was programmed to load a 100 pM library and sequenced in CLR mode on a PacBio SMRTcell 8M, with a Sequencing Plate 2.0 (Pacific Biosciences, Menlo Park, CA, USA) and 2 h of pre-extension time and acquiring one movie of 15 h per SMRTcell.

Whole-genomic DNA of the wild-type strain, the 10th, 25th, 50th, and 75th generations and the three isolated clones were sequenced with the Ion S5 system on the Get-Biopuces sequencing platform (Toulouse, France, https://get.genotoul.fr, accessed on 20 March 2021).

### 2.7. Functional Annotation and Genome Analysis

Gene prediction and functional annotation of the wild-type strain of *E. limosum* B2 were performed using the NCBI Prokaryotic Genome Annotation Pipeline PGAP [24]. The genome sequence can be found under the accession number CP097376.1. Whole-genome sequence alignment of *E. limosum* B2 against *E. limosum* ATCC 8486 was performed using ProgressiveMauve software (v2.4.0) [25].

### 2.8. Methylome and Type I Restriction-Modification System Analysis

For analysis of the whole DNA methylation level, methylation of adenine on the 6th carbon (m6a) and methylation of cytosine on the 4th carbon (m4c) were determined using the ipdSummary and Motifmaker tools included in the SMRT Link software (v10.1). The protein sequence alignment for the analysis of the type I RM system was performed using ClustalW in Ugene software (v40.1). Conserved domain analysis was performed using the NCBI conserved domain search tool.

### 2.9. Protein Isolation and Total Proteome Analysis

Wild-type *E. limosum* B2 and clone 2 were cultured in methanol mineral medium containing 0.5 g/L yeast extract in duplicate. Cells were harvested at mid-exponential growth phase, and sample volumes were adjusted to normalize the cell number to 10^9^ for each condition. The samples were then centrifuged at 16,000× *g* for 30 s, and the supernatants were discarded. Cell pellets were washed twice with 1 mL of phosphate-buffered saline solution. After discarding the supernatant, the cell pellets were frozen in liquid nitrogen and stored at −20 °C. Total proteins were isolated using the SPEED extraction method [26]. Each pellet was resuspended in 150 µL of trifluoroacetic acid (TFA, Supelco MS Grade, Darmstadt, Germany), warmed at 70 °C for 3 min and centrifuged at 16,000× *g* for 1 min. A total of 130 µL of supernatant was carefully transferred to a fresh 1.5 mL Eppendorf tube. Fourteen microliters of supernatant were taken for each sample and diluted in 56 µL 75% TFA to measure the protein concentration by the Bradford assay. Each sample was then neutralized with 1.5 mL of TRIS 2 M, prepared in LC–MS grade water (Supelco), transferred to glass vials and freeze dried. Samples were sent to the John van Geest Cancer Research Center at Nottingham, resolubilized, purified and injected into a TripleTOF 6600 mass spectrometer (Sciex). Two injections per sample were performed, giving a technical duplicate for each sample. Sequential window acquisition of all theoretical spectra (SWATH) and additional data treatment using DIA-NN (https://github.com/vdemichev/DiaNN, accessed on 1 April 2022) were performed to analyse the two groups of samples [27].

## 3. Results

### 3.1. De Novo Sequencing, Annotation and Analysis of the E. limosum B2 Genome

Although the genomes of several *E. limosum* strains have been sequenced, the genome sequence of *E. limosum* B2, the most well-studied strain from a physiological point of view, remains unknown [28,29,30]. De novo sequencing was performed on the WT strain using single molecule real-time technology (PacBio). A total of 76,620 long reads were used to produce two linear contigs of 4.06 Mb and 358 kb for a total genome size of 4.424 Mb. The main issue in obtaining one circular contig was the presence of a prophage, present in two copies in the genome and localized at one end of each of the contigs. A complete circular assembly of 4,421,327 base pairs was created using long-range PCR associated with Sanger sequencing. Based on large reads, PacBio sequencing technology can be prone to errors [31]. To correct possible errors from the sequencer, the circular genome sequence was compared to the whole genome sequence of the *E. limosum* B2 WT strain obtained by the IonTorrent S5 sequencer. One nucleotide insertion was observed in the PacBio sequence, localized in an *ompD* homologous gene, which was not confirmed by Ion S5 sequencing.

The genome sequence of *E. limosum* B2 has a G+C content of 47.2% and contains 4192 genes, 51 tRNA genes, 16 rRNA genes including 5 23S, 16S and 5S rRNA operons, a supplementary 5S rRNA gene, 1 tmRNA and two clusters of CRISPR genes, similar to the *E. limosum* SA11 and KIST612 strains [28,29]. The genome of *E. limosum* B2 showed 99.837% sequence identity with the *E. limosum* ATCC 8486 strain.

### 3.2. Genomic Comparison of E. limosum B2 vs. E. limosum ATCC 8486

Differences between genes and intergenic regions were identified (Table 1).

In addition to large genomic rearrangement, as we describe later, a total of 21 small genomic differences, including 12 single nucleotide polymorphisms (SNPs), six nucleotide insertions and three single nucleotide deletions, were observed for the *E. limosum* B2 genome compared to the *E. limosum* ATCC 8486 chromosome. Differences occurred mainly in genes coding for regulators, such as a potential response regulator (M5595_11305) introducing an Arg^63^Gly substitution in the protein or a transcriptional activator (M5595_16695) introducing a silent mutation. A total of four genetic differences were detected in genes coding for a prophage operon, one in a terminase encoding gene (M5595_17445) leading to an Ala^421^Glu substitution in the protein, two in intergenic regions between genes of unknown function and one in a gene coding for a hypothetical protein. Furthermore, three insertions were noticed in a gene of *E. limosum* B2 encoding a potential S subunit (M5595_00360) of a type I restriction endonuclease. Other genomic differences were localized in genes encoding transporters, such as a uric acid permease-encoding gene (M5595_04320) or a bacitracin export permease-encoding gene (M5595_08425). Additionally, deletion of an insertion sequence (IS) was detected in the genome of *E. limosum* B2 compared to *E. limosum* ATCC 8486. This IS was present in six and five copies in *E. limosum* ATCC 8486 and B2 strains, respectively. The missing IS in *E. limosum* B2 was located upstream of a sodium/phosphate transporter-encoding gene, reducing the size of the gene by 42 nucleotides in the *E. limosum* ATCC 8486 gene according to the annotation. Whole genome alignment using ProgressiveMauve software also revealed genome reorganization of five locally colinear blocks (LCBs) with reverse complement orientation (Figure 1).

### 3.3. Whole DNA Methylation Level

The DNA methylation of the chromosome is an epigenetic modification that plays a crucial role in the response to environmental stresses, genome structure rearrangements and gene regulation [32,33]. To explore the methylome of *E. limosum* B2, a kinetic data analysis was performed on the SMRT sequence obtained on the PacBio system to identify the locations of m6A and m4C methylated nucleotides. The five restriction-modification (RM) systems identified in *E. limosum* ATCC 8486, including one type I RM and four type II RMs, were also present in the *E. limosum* B2 strain. Methylome data from *E. limosum* B2 showed only three different motifs of the m6A type for a total of 3974 methylated motifs out of 8444 probable motifs. The *E. limosum* B2 strain had a modification frequency of 0.96 modifications/kb (Table 2).

The GCGRA^m6^G motif was most prevalent, with 2058 occurrences among a total of 3042 sites. A GCGRA ^m6^G-like motif, ‘GCGCA ^m6^G’, was identified in *Rhuminococcus flavefaciens* FD-1 according to the REbase databank. This motif site is recognized by a type II RM (RflFIII), sharing 66% similarity to Class I SAM-dependent methyltransferase from *Eubacterium* species but only 21% similarity with N-6 DNA methylase from *E. limosum* B2.

Compared to the methylome of a close metabolic neighbour, *A. woodii,* almost all the detected motifs carried a methyl group, while the motif fraction that contained methylated nucleotides was lower (67.6%) for *E. limosum* B2 [34]. Furthermore, the number of methylated nucleotides expressed per kb of DNA was calculated to be 0.96 modifications/kb for *A. woodii*. The CAAAAA ^m6^R methylated motif present in the *E. limosum* B2 genome was also detected in *A.*
*woodii* [33]. This adenine in position 6 in the motif is methylated by a type II methyltransferase, annotated N-6 DNA methylase (Awo_c14460), sharing 48.92% similarity to the N-6 DNA methylase of *E. limosum* B2 (M5595_07210). A total of 27.9% of CNNTAYNNNNNTCC carry a modification of their adenine at positions 5. Although the associated enzyme remains unknown, these nonpalindromic bipartite motifs are likely recognized by a Type I RM system and most likely encoded by the M5595_00345 to M5595_00360 genes.

### 3.4. Strain Adaptation on Methanol-Defined Medium without Yeast Extract

The first step for growth optimization on methanol substrate was to adapt the strain on a YE-free defined medium. The WT strain, initially cultivated in M187-rich medium, was inoculated in methanol-defined medium without yeast extract to obtain a strain that could grow in YE-depleted medium. Two subcultures in freshly defined medium were performed to ensure YE from the original rich medium culture was at least diluted 400-fold and played a minor role in the YE-depleted medium. The cell density of the WT strain in methanol-defined medium was compared to that in methanol-rich medium as a control (Figure 2).

After a 3-day lag phase, a population started to grow, showing a maximum OD_620_ of 0.670 ± 0.05 and a maximum specific growth rate (µmax) of 0.017 ± 0.002 h^−^^1^. The WT strain cultivated on M187 rich medium showed a maximum OD_620_ of 2.165 ± 0.05 with a µmax of 0.049 ± 0.003 h^−^^1^. The growth rate in methanol-defined medium decreased by 65% compared to that in the rich medium, but the WT strain showed its ability to overcome YE depletion. Obtaining a strain that can grow without YE represents a strong asset to precisely determine metabolic fluxes from different carbon sources using a genome-scale model. However, the specific growth rate of 0.017 h^−^^1^ remained low compared to the maximum growth rate of 0.05 h^−^^1^ measured for the *E. limosum* B2 strain in semi-defined medium supplemented with YE 0.5 g·L^−^^1^ YE [10]. The next step of the study was to improve the physiological characteristics of the strain in methanol-defined medium by ALE.

### 3.5. Adaptative Laboratory Evolution of E. limosum B2 in a Methanol-Defined Medium

The ALE on *E. limosum* ATCC 8486 was reported on carbon monoxide in semi-defined medium supplemented with 2 g·L^−1^ YE. The results showed a notable improvement in the physiological characteristics of the *E. limosum* ATCC 8486 strain, with a maximal optical density (620 nm) and a growth rate that increased 2.14-fold and 1.44-fold [35], respectively. To the best of our knowledge, no ALE has been performed using methanol as a substrate with *Eubacterium* species. Few studies have focused on ALE to improve growth on methanol. Such an experiment was performed on the natural methylotroph *Saccharomyces cerevisiae,* improving the final biomass by 44% after 230 generations [36]. Additionally, ALE on methanol was performed for synthetic methylotrophs such as *Escherichia coli* or *Corynebacterium glutamicum* to improve their tolerance [37,38] to methanol. The ALE was performed on *E. limosum* B2 to obtain a strain with the best growth performance on methanol-defined medium. In total, the strain was cultured for more than 100 generations with a subculture performed every three generations. The ALE showed a clear improvement in the maximal growth rate (Figure 3A).

The lag phase was gradually reduced as the strain was subcultured, from approximately 70 h for the first generation to less than 5 h for the evolved population. The growth rate progressively increased over generations, with a µmax at 0.022 ± 0.001 h^−1^ for the 10th generation, 0.047 ± 0.0084 h^−1^ for the 25th generation, and 0.057 ± 0.0026 h^−1^ for the 50th generation, reaching a maximal value of 0.076 ± 0.0019 h^−1^ for the 75th generation. Beyond this last generation, no significant physiological growth improvement was noticed, suggesting that the population reached its maximal potential. In comparison, a maximum growth rate of 0.07 h^−1^ was observed for *E. limosum* B2 in semi-defined methanol (200 mM) medium supplemented with sodium acetate at 12.7 mM [10].

To further investigate the physiologic parameters of the strain on methanol-defined medium, product measurements were systematically performed before each subculture, and values are presented for the 10th, 25th, 50th and 75th generations (Figure 3B). Significant changes in carbon distribution were observed for acetate production yield between generations, with a significant yield increase of approximately 80% observed between the 10th and 50th generations, from 45 ± 4 to 77 ± 5 mC acetate/100 mC MetOH. The mean acetate yield decreased by approximately 20% between the 50th and 75th generations, reaching 50.5 ± 4.5. In contrast, the butyrate production yield showed a significant decrease of approximately 40% between the 10th and 50th generations from 67 ± 2 to 39.5 ± 0.5 mC butyrate/100 mC MetOH. For the 75th generation, the mean butyrate yield showed an increase to 55.5 ± 7.5 mC butyrate/100 mC MetOH, approximately 40% that for to the 50th generation. The carbon flux toward acetate tended to increase from the 10th to 50th generation, while the reverse phenomenon was observed for butyrate production. Carbon from methanol tended to have an equimolar distribution between acetate and butyrate for the 75th generation. Globally, the ALE did not drastically affect the carbon balance between the two products. Carbon incorporation into biomass rapidly increased from the 10th generation (7.5 ± 0.5) to reach 28 ± 1 mC biomass/100 mC MetOH for the 75th generation.

According to the statistical analysis method applied, no significant change was noticed over generations for the CO_2_ consumed, which is a necessary cosubstrate for methylotrophic growth of *E. limosum* B2 [10]. In general, more carbon from the substrate was dedicated to biomass production through ALE, while the CO_2_ consumption rate remained constant. The evolved population on synthetic methanol medium tended to have an equimolar carbon distribution from methanol and CO_2_ to produce acetate and butyrate.

### 3.6. Isolation of Individual Clones Growing on Methanol Mineral Medium

The next step of the study was to isolate a strain growing on a methanol-defined medium in the absence of cysteine, i.e., a methanol mineral medium (MMM). To obtain a strain able to grow on a defined medium without cysteine and with methanol as substrate, cells from the advanced evolved population (75th generation) were first spread on a solid glucose mineral medium (SGMM) as no colonies were obtained when spread on a solid MMM (SMMM) supplemented or not with cysteine. This phenomenon was probably due to CO_2_ accessibility issues (although cultures were performed in an anaerobic chamber with 5% CO_2_) along with possible methanol evaporation. After growth on SGMM, three colonies were selected and cultured in MMM medium at 37 °C. However, after three subcultures, clone growth declined progressively, presumably from cysteine deficiency. Although *E. limosum* B2 was not documented as auxotrophic for cysteine, alternative sources of sulphur were tested to overcome this presumed auxotrophy [39]. Sodium dithionite, sodium metabisulfite and sodium thiosulfate were added separately to the MMM at 1, 2 and 4 mM. Clones grew on all sulphur sources, with a better maximal biomass observed at 4 mM sodium dithionite. Although sodium dithionite slightly improved growth performance compared to sodium thiosulfate at 4 mM, this last sulphur source was chosen for two reasons: (i) it allows similar growth performance compared to a methanol-defined medium with cysteine and, (ii) sodium dithionite is also a reducing agent. Therefore, we chose to dissociate the source of sulphur from the source of the reducing agent (titanium citrate at 2 mM) to obtain more reproducible fermentation data.

### 3.7. Growth Profile of Adapted Clones in Liquid MMM

Growth profiles of the three isolated clones were determined in liquid MMM supplemented with sodium thiosulfate. Optical density monitoring of the three clones showed similar growth profiles, with a maximum OD of 3.69 ± 0.43 for clone 1, 4.03 ± 0.08 for clone 2 and 3.44 ± 0.37 for clone 3. The maximum growth rate was 0.054 ± 0.008 h^−^^1^ for clone 1, 0.060 ± 0.003 h^−^^1^ for clone 2 and 0.047 ± 0.004 h^−^^1^ for clone 3 (Figure 4A). Globally, clone 2 showed the best growth performance compared to the two others.

Analysis of the carbon balance for the three clones showed that the acetate yield of clone 3, with a value of 48 ± 6, was significantly lower than those of clones 1 and 2, with values of 60.5 ± 4.5 and 59.7 ± 4.16 mC acetate/100 mC MetOH, respectively (Figure 4B). The butyrate yield of clone 1, with a value of 37.5 ± 6.5, was significantly lower than those of Clones 2 and 3, with values of 43 ± 4 and 51 ± 8 mC butyrate/100 mC MetOH, respectively. No significant difference was observed among the clones for biomass yield. However, CO_2_ consumption was significantly higher for clone 2, with a value of 23.27 ± 1.15 mC CO_2_ consumed/100 mC MetOH, compared to clone 1 and clone 3, with values of 12 ± 3 mC and 18.5 ± 0.5 CO_2_ consumed/100 mC MetOH, respectively.

Culture of the wild-type strain showed no significant growth on MMM with sodium thiosulfate addition, indicating that genetic modification/regulation could be responsible for the ability of the strain to grow on MMM. Whole-genome sequencing of the 10th, 25th, 50th and 75th generations from the evolved population and the 3 isolated clones was performed to potentially identify mutated genes responsible for (i) the improved growth on MMM plus cysteine during ALE and (ii) the potential ability of the isolated clones to grow on MMM supplemented with thiosulfate.

### 3.8. Mutation Profiles of the ALE Lineage by Whole Genome Resequencing

To better understand the underlying mechanisms of growth improvement on MMM supplemented with cysteine, the genome sequences of the evolved populations after the 10th, 25th, 50th, and 75th generations were determined. Furthermore, the genomes of the three selected clones growing on liquid MMM supplemented with thiosulfate were also sequenced (Table 3).

A total of 28 genetic modifications, including two conserved mutations, were identified by comparison with the wild-type strain sequence. Surprisingly, no mutation occurred in genes directly involved in methanol metabolism or the WLP. Two conserved one-base insertions spaced by only 69 bp were found in all generations and clones (except for one insertion that was not observed in clone 3) at the end and the beginning of two adjacent genes (coding for hypothetical proteins) and part of a large prophage operon. Such conserved mutations over generations suggest a response by genomic adaptation of the strain to the restrictive growth conditions on methanol-defined medium without YE. However, understanding the role of insertions in genes coding for hypothetical proteins remains difficult. An ALE experiment performed to improve the growth of the *Sporomusa ovata* acetogen on a liquid methanol mineral medium also showed similar insertions in genes belonging to a prophage operon [40]. In addition to the mutations in genes from a prophage operon, the structure of type I RM encoding genes of the three adapted clones was different compared to the WT strain (Figure 5).

The system is composed of one R subunit, one M subunit, one integrase and three S subunits. Among the S subunits annotated, two subunits seem truncated (M5595_00355 and M5595_00360), while one was complete (M5595_00345). We found four repeated sequences, including two repeated sequences in the complete S subunit and two other sequences in truncated S subunits in the WT and adapted strains. Interestingly, we observed a homologous recombination event without mutation in these S subunits between the adapted and WT strains, giving a complete S subunit in the truncated S subunit region and vice versa. The apparent complete S subunit encoding the gene sequence was 1149 bp for the WT strain versus 1236 bp for the clones 1, 2 and 3, indicating a different protein sequence. After alignment of protein sequences of both S subunits of WT and adapted strains, the structure appeared to be different at the C-terminal side of the protein (Figure 6). Furthermore, the search for conserved domains indicated two target recognition domains (TRDs), the first located in the 101–192 residue interval, which corresponds to the modified region, and the second in the 236–396 residue interval. The sequence analysis of the S subunit from the adapted strains strongly suggests the recognition of a different DNA sequence. This was confirmed by the determination of the methylome of the clone 2 (Table 2) that non only present a different recognition sequence for the type I RM (CNNTAYNNNNGTG instead of CNNTAYNNNNNTCC) but also a much higher proportion of CAAAAA ^m6^R methylated motifs (83.7 versus 37.3%).

Eleven mutations were detected in the 10th generation compared to the WT strain, with eight mutations at 100% frequency and three other mutations below 58.9% frequency. The only SNP detected in a gene coding for a metabolic enzyme was in the gene encoding the butyryl-CoA: acetate CoA transferase, but this mutation was not detected in the next generations, indicating that it was not needed to improve growth on methanol. Globally, the mutations found in early generations declined for more advanced evolved populations. Furthermore, the three isolated clones showed only a few mutations compared to the WT strain and shared no mutation in common except the ones in the prophage operon and the reorganization of the S subunit encoding genes of the type I RM.

Whole genome sequencing for the 10th, 25th, 50th, and 75th generations and isolated clones during the ALE showed no evidence of genetic modification in genes involved in methanol oxidation or in genes coding for the WLP and enzymes for energy conservation. However, the WT strain could grow in liquid methanol mineral medium without YE, while the evolved clone 2 could grow if sodium thiosulfate was added. To characterize the effect of the modified methylome on gene expression and also better understand the adaptation mechanisms on synthetic medium, the complete proteome profile was performed for WT and clone 2 strains.

### 3.9. Proteomic Analysis of the WT and Evolved Clone 2

#### 3.9.1. Global Analysis of the Proteomes

The total proteome of *E. limosum* B2 was compared between the WT strain and clone 2 on MMM supplemented with 0.5 g/L YE to sustain the growth of the WT strain, as it failed to grow on MMM supplemented with sulphur as the only mineral source. The optimal cell density remained similar for both strains, with an OD_620_ of 4.0. However, the specific growth rate of clone 2 reached 0.084 ± 0.005 h^−1,^ compared to 0.035 ± 0.001 h^−1^ for the WT strain, improving the maximal growth rate 2.4-fold for the adapted strain compared to the WT strain (Figure 7). Although the MMM was supplemented with YE, the clone 2 strain showed a significant improvement in growth compared to the native strain.

To evaluate the efficiency of total proteome extraction, the abundance of ribosomal proteins was analysed as a representative of cytosolic proteins, while the extraction of membrane-bound proteins was assessed by examining the proteins of the ATP synthase complex. A total of 46 ribosomal proteins out of 51 were identified for both conditions, indicating efficient extraction using the SPEED method with an average coefficient of variation of 11.5% for the clone 2 strain and 35% for WT (Figure 8A). The variability among cytosolic proteins between replicates for the native strain was superior to that for the adapted strain. This difference might be explained by a different biomass composition between strains. After cell lysis in trifluoroacetic acid (TFA), a larger amount of precipitated compound was observed for the native strain compared to the adapted strain. Based on the ratio of the total ribosomal intensity value to the total proteome intensity value for both conditions, the relative number of ribosomes per cell was 2.57-fold higher in the clone 2 strain than in the native strain. This value was closely related to the specific growth rate difference of both strains, as the clone 2 strain showed a specific growth rate 2.4-fold higher than that of the WT strain.

Regarding the efficiency of the total protein extraction method for membrane-bound proteins, seven out of nine constitutive proteins of the ATP synthase complex were identified. The ATP synthase in *E. callanderi* KIST 612 was identified to be a unique combination of a Na^+^ A_1_A_0_ ATP synthase with a V-type *c* subunit organized in an operon of nine genes [41]. A similar operon was found in the genome of *E. limosum* B2, and seven out of nine constitutive proteins of the ATPase operon were identified by mass spectrometry (Figure 8B). The ATPase complex is organized as an A_1_A_0_ ATP synthase with a catalytic head constituted by A and B subunits, a central stalk comprising the cytoplasmic subunits C, D and F and a rotor located in the membrane including eight to ten *c* and *a* subunits that are stoichiometrically equivalent. In addition, A-type ATP synthases contain subunits H and E [42]. The precise stoichiometry for the ATP synthase of *E. limosum* B2 is currently unknown and requires further investigation using imaging technology to determine the structure of the complex [43]. For example, the stoichiometry of *Pyrococcus furiosus* was established with the following formula: A_3_B_3_CDE_2_FH_2_*ac*_10_ [44]. Based on the previous stochiometric formula, the intensity value measured for the different subunits of the ATP synthase complex did not match. A clear difference in the proportion of total proteome intensity was observed between the *a* and *c* subunits, the C and D subunits, and the A and B subunits. The extraction method applied to the samples did not allow total membrane-bound protein extraction but allowed the identification of a sufficient proportion of membranous proteins, as numerous transporters were identified through this extraction method coupled to mass spectrometry.

By combining replicates of both strain cultures, a total of 1256 different proteins were detected, with 1197 proteins identified for the clone 2 strain and 1015 proteins for the WT. A difference of 182 supplementary proteins was observed for the clone 2 strain, suggesting that the mobilization of higher protein diversity led to better growth performance on methanol synthetic medium. The complete list of differentially produced proteins is available in Appendix A.

The distribution of differentially produced proteins between the clone 2 strain and native strain (One-sided Student’s *t*-test, *p* value < 0.05) showed over- and underproduced proteins with more than or less than 1 Log_2_ fold change. The global proteome showed substantial differences between both conditions, with a total of 91 significantly differentially produced proteins with a fold change (FC) value >2 or <−2 and a *p* value < 0.05. A total of 52 proteins were overproduced, while 39 proteins were underproduced in the clone 2 strain compared to the WT strain (Figure 9). Proteins from various metabolic functions and pathways were identified as differentially expressed between the conditions, suggesting metabolic flux reorganization between the two strains.

#### 3.9.2. Analysis of the Methanol Oxidation and Wood–Ljungdahl Pathways

Methanol oxidation has been well characterized in acetogens with evidence of reverse use of the methyl branch of WLP in *E*. *callanderi* KIST612, a closely related species, to sustain reducing equivalent equilibrium [16]. Additionally, energy production and reducing equivalent balance were reported to be ensured by the Na^+^-dependent Rnf complex associated with the ATP synthase complex, creating a proton motive force by importing Na^+^ ions to oxidize NADH and reduce Fd [15,16]. Before entering the WLP, methanol is oxidized and coupled to a tetrahydrofolate (THF) molecule by a methanol-specific methyltransferase complex composed of three proteins containing two catalytic domains [45]. Methanol corrinoid methyltransferase (MTI) transfers the methyl group of methanol to the central cobalt atom of corrinoid protein (CoP), followed by the transfer of the methyl group to a THF molecule by methyltetrahydrofolate cobalamin methyltransferase (MTII). No mutation occurred in the central metabolism coupled to the energy conservation system, and as expected, the proteomic analysis revealed a high abundance of proteins involved in methanol oxidation and the WLP for both conditions, but although the level of all these proteins was higher in the evolved clone, only two were above the 2-fold difference that we set as a threshold (Figure 10). Carbon monoxide dehydrogenase (CooS) was significantly more abundant in the adapted strain, showing an FC value of 2.22, as well as the thiolase, FC value of 2.08, involved in butyrogenesis. Globally, proteins of the CODH/ACS complex in the carbonyl branch, which are involved in CO_2_ fixation into acetyl-CoA, and proteins of the butyrogenesis pathway were slightly overproduced compared to proteins involved in the methyl branch and acetogenesis.

#### 3.9.3. Analysis of Gluconeogenesis and Anaplerotic Reactions

When glucose is not available during autotrophic or methylotrophic conditions, acetyl-CoA, CO_2_, ATP and reducing power are used in gluconeogenic reactions to produce all the intermediates needed for nucleic acid, amino acid, cell wall and essential cofactor biosynthesis [46]. The first step is catalysed by the pyruvate-ferredoxin oxidoreductase (PFOR) A0A317RSE4 to produce pyruvate [47] from acetyl-CoA, reduced ferredoxin and CO_2_ (Figure 11).

Interestingly, substantial overproduction of PFOR was observed, showing an FC value of 5.23, and this was among the most abundant differentially produced proteins detected. Another gluconeogenic enzyme, glucose-6-phosphate isomerase (PGI), was the most overproduced protein, with an FC value of 7.60. Glucose 6-phosphate was demonstrated to represent a key metabolite for peptidoglycan as well as cell wall polysaccharide synthesis. Such upregulation of both the PFOR- and PGI-encoding genes must be a solution developed by the evolved strain to improve its growth capacity in synthetic medium. The carbon recovered in biomass for the clone 2 strain represented approximately 20% of the total carbon metabolized during methylotrophic fermentation against approximately 6.5% for the 10th generation population (Figure 4 and Figure 5).

The pyruvate carboxylase (PC), which catalyses the carboxylation of pyruvate into oxaloacetate, the first metabolic intermediary of the TCA cycle, was underproduced (FC value −2.29). Furthermore, the 2-oxoglutarate ferredoxin oxidoreductase subunits alpha and beta (KGOR) were the most highly underproduced proteins (FC values −17.11 and −19.47, respectively). As isocitrate dehydrogenase was upregulated (FC 3.11), this might allow a high flux of alpha-ketoglutarate production that could further be used for glutamate production.

#### 3.9.4. Sulphate Metabolism for the Biosynthesis of Sulphur-Containing Amino Acids

The clone 2 strain can grow on methanol mineral medium with the addition of sodium thiosulfate instead of cysteine, which is required for the WT strain. Interestingly, anaerobic sulphite reductase subunits A, B and C (A0A317RV53, A0A317RVB5 and A0A317RTV7) were among the most statistically overproduced proteins in the adapted clone 2, showing an average FC value of 4.47. In mycobacteria, anaerobic sulphite reductase has previously been demonstrated to be essential for growth on sulphate or sulphite as the sole source of sulphur [48]. Anaerobic sulphite reductase catalyses sulphide (S^2−^) production from sulphite (SO_3_^2−^). Cysteine is then synthesized from sulphide and serine by a two-step enzymatic pathway [49]. First, serine is acetylated by a serine transacetylase (CysE) to form O-acetylserine, and second, cysteine is produced from O-acetylserine and sulphide via a reaction catalysed by a cysteine synthase (CysK). Furthermore, CysK (A0A317RWJ0) was overexpressed in the clone 2 strain with an FC value of 1.87. There remains some uncertainty regarding the sulphite supply by thiosulfate. Sulphite can be produced either by the reduction of sulphate (SO_4_^2−^) or by the reduction of thiosulfate by a thiosulfate reductase [50]. However, no thiosulfate reductase-encoding gene was found in the genome of *E. limosum* B2 or among the detected proteins, and SO_4_^2−^ from iron sulphate did not sustain the growth of the strain in synthetic medium. Where does the sulphite come from if sulphate is not the sulphur source? Evidence of cysteine production with thiosulfate was reported in *E. coli* [51]: L-serine can be acetylated to O-acetylserine by a serine acetyltransferase, and the product can be further converted with the addition of thiosulfate to S-sulphocysteine by an O-acetyl-homoserine sulphhydrylase OASS (A0A317RPB8). Then, L-cysteine can be synthesized from S-sulphocysteine by thioredoxin/glutaredoxin, liberating sulphite, which would be recycled for further cysteine production. The OASS was highly overproduced in the adapted strain (FC 4.30), suggesting the utilization of this pathway to produce cysteine in the adapted clone 2 (Figure 12).

Interestingly, 5-methyltetrahydrofolate-homocysteine methyltransferase, also known as methionine synthase, was the second strongest underproduced protein (FC −3.56) among amino acid biosynthesis-related proteins. This enzyme was shown to have methionine synthase activity [52]. Methionine synthase is involved in the last step of methionine biosynthesis, using homocysteine and methyl-cobalamin as cofactors. Decreased production of methionine synthase could suggest the preservation of homocysteine to produce cysteine via the transsulfuration pathway [53]. In addition to cysteine production through L-serine transformation and thiosulfate utilization, this pathway was probably used by the clone 2 strain to produce cysteine. This function likely emerged during the evolution of the strain, as cystathionine gamma-synthase (A0A317RYT5) and cystathionine beta-lyase (A0A317RVG5), two key enzymes of the transsulfuration pathway, were only detected in the clone 2 strain.

#### 3.9.5. Other Differentially Produced Proteins

Because all ALE processes were performed without YE, the evolved strain strongly regulated its metabolism to produce the required amino acids. While proteins involved in amino acid biosynthesis were found to be slightly overproduced, others were strongly underproduced, such as leucine dehydrogenase (A0A1H0LY56), with an FC value of −13.81.

Surprisingly, several enzymes involved in propanediol utilization (*pdu* operon) were strongly underproduced in clone 2, indicating an important function in the native strain. The enzyme BMC domain-containing protein (A0A0U3FC86) which is involved in the bacterial microcompartment for 1,2 propanediol (1,2 PDO) or glycerol metabolism exhibits significant underproduction (FC −6.76). The enclosed proteinaceous structure metabolizes 1,2-propanediol to propanol and propionate or to produce 3-hydroxypropionate from glycerol. Encapsulation is necessary to limit cytosolic exposure to propionaldehyde, a toxic metabolic intermediary [54]. The presence of cytosolic microcompartments was described for *E. maltosivorans* with 1,2 PDO or betaine as carbon sources. Furthermore, BMC structures were also reported on *A. woodii*, a close relation, with 1,2 PDO and 2,3 butanediol as carbon sources, but no BMC was detected with methanol as a carbon source. The presence of proteins from the *pdu* operon at high intensity for the native strain remains enigmatic, as no propanol or 3-hydroxypropionate was detected by HPLC.

## 4. Discussion

In this study, the genome of the most studied (from a physiological point of view) strain of *E. limosum* was sequenced. The *E. limosum* B2 genome shows 99.837% sequence identity with that of the *E. limosum* ATCC 8486 strain. Although the two genomes are very similar, they exhibit chromosomal rearrangement in prophage operons, insertion sequences and 5 colinear blocks (LCBs) with a reverse complement orientation. Furthermore, several differences were found in genes coding for regulators and an S subunit-encoding gene of a type I RM system.

*Eubacterium limosum* B2 required unknown components in the complex media for growth on methanol [10]. To overcome this problem, we applied an adaptive evolution approach to develop an optimized strain of *E. limosum* B2 capable of growth on a methanol-defined medium without yeast extract. By this approach, the growth rate was enhanced 3.45-fold in 75 generations. From the evolved population, three clones were isolated and grown on methanol mineral medium without cysteine (which was replaced by sodium thiosulfate). Among the three isolated clones, the clone 2 strain showed the best growth characteristics, showing a maximal growth rate 2.72-fold higher than that of the 10th generation population cultured in methanol-defined medium supplemented with cysteine. Genome sequencing of the WT, 10th, 25th, 50th, and 75th generations and three evolved clones showed no specific mutation in central carbon metabolism or conserved mutation over generations except in a large phage operon and in the S subunit encoding genes of a type I RM system. Proteomics data, on the other hand, showed important differences in the proteomic compositions of the WT and clone 2. First, overproduction was observed for the enzymes of both (i) the carbonyl branch of the WLP and (ii) the butyrogenesis pathway. This should lead to an improvement in CO_2_ fixation ability by both (i) an enhancement in reduced ferredoxin production by the electron bifurcating complex Bcd-Etf in the butyrogenesis pathway and (ii) by an increase in the level if CODH that also requires reduced Fd^2−^ as a source of electrons.

Better growth of clone 2 could also be the result of higher production of enzymes involved in the synthesis of key building blocks of the cells, such as the PFOR involved in pyruvate production from acetyl-CoA, PGI responsible for glucose-6-P production from fructose-6-P or IDH involved in a-keto-glutarate production needed for glutamate production. The ability of clone 2 to grow on a methanol mineral medium in the absence of cysteine could be the result of a combination of higher production of (i) the anaerobic sulphite reductase and (ii) the enzymes of the transsulphuration pathway allowing cysteine biosynthesis from thiosulfate. The considerable modification of the proteomic profile of the evolved clone 2 was associated to changes of its methylome profile suggesting that it is the result of an epigenetic phenomenon. The rearranged S subunit of the type I RM in the evolved clones, recognize a different motif (CNNTAYNNNNGTG instead of CNNTAYNNNNNTCC) and a higher proportion of these motifs are methylated. Similarly, a much higher proportion of type II motifs (CAAAAA ^m6^R) are also methylated (83.7 versus 37.3% in wild type). These modifications in the methylome profile are probably responsible in the large change in gene expression. A similar phenomenon was previously demonstrated in *Bacillus pumilus* when the type I RM-encoding genes were deleted [55]. This study highlighted the metabolic/genetic adaptation of the *E. limosum* B2 strain growing on methanol-defined medium and showed the importance of improving our knowledge of cell adaptation mechanisms in a constrained environment. The evolved clone 2 that grows on a mineral medium should allow us to develop and use a genome-scale model of *E. limosum* B2 for both flux analysis and rational metabolic engineering of *E. limosum* B2, as previously performed with *Clostridium acetobutylicum* [56]. Furthermore, to better characterize the metabolism of this strain, it can now be used in chemostat cultures in mineral medium at a constant specific growth rate using different carbon sources to perform a complete systems biology analysis.

## Figures and Tables

**Figure 1 microorganisms-10-01790-f001:**
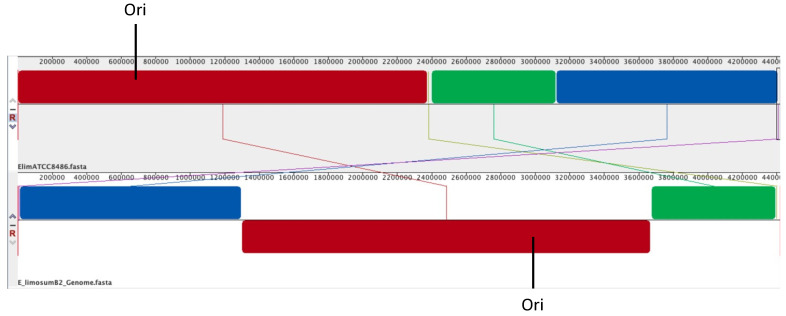
Whole genome alignment of *E. limosum* B2 (bottom) to *E. limosum* ATCC 8486 (top) using ProgressiveMauve software [25]. Ori, origin of replication at genomic position 689,833 for *E. limosum* ATCC8486 and 2,982,134 for *E. limosum* B2.

**Figure 2 microorganisms-10-01790-f002:**
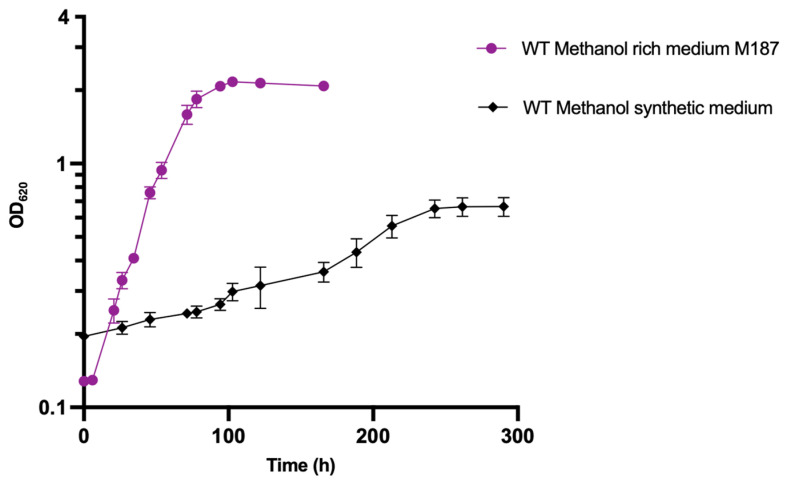
Growth profile of *E. limosum* B2 on methanol-rich and synthetic medium without YE. Cultures were performed in triplicate and errors bars indicates standard deviation.

**Figure 3 microorganisms-10-01790-f003:**
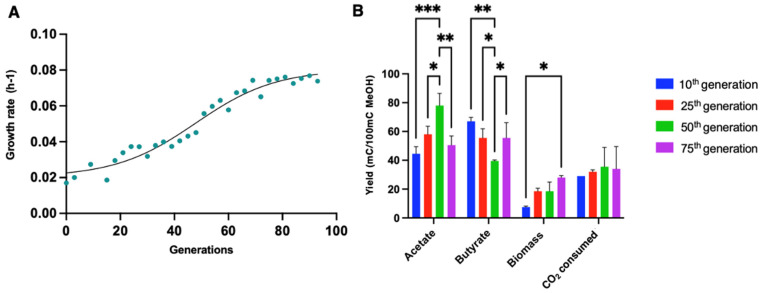
Evolution of physiologic characteristics during ALE on methanol methylotrophic growth of *E. limosum* B2. (**A**) Evolution of the maximum growth rate for each generation during ALE. (**B**) Carbon yields obtained from consumed methanol to acetate, butyrate, biomass, and CO_2_. Yields are expressed in millimolar of carbon (mC) per 100 mC of methanol consumed. Cultures were performed in triplicate and error bars indicates standard deviation. Asterisk indicates that the results were significantly different from the control according to the one-sided Student’s *t*-test: *: *p* value ≤ 0.05, **: *p* value ≤ 0.01, ***: *p* value ≤ 0.001.

**Figure 4 microorganisms-10-01790-f004:**
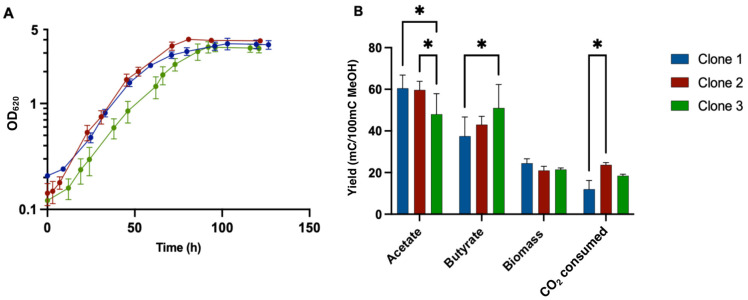
Physiological study of isolated clones from evolved populations growing on YE and cysteine-free synthetic medium with 200 mM methanol. (**A**) Growth profile monitoring by measuring the optical density (620 nm). (**B**) Carbon balance from methanol to products, biomass and CO_2_ consumed. Cultures were performed in triplicate and error bars indicates standard deviation. Asterisk indicates that the results were significantly different from the control according to the one-sided Student’s *t*-test: *: *p* value ≤ 0.05.

**Figure 5 microorganisms-10-01790-f005:**
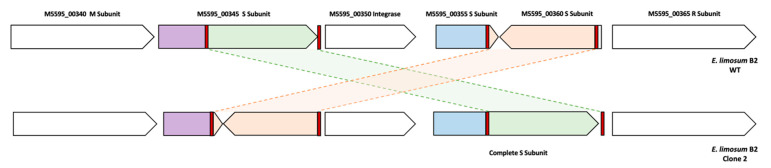
Restriction-modification system type I organization in *E. limosum* B2 WT and clone 2. Repeated sequences are highlighted in red while homologous recombination is represented by interchange of green (714 bp) and orange (642 bp) genomic parts. The organization of this region in clones 1 and 3 was the same as clone 2.

**Figure 6 microorganisms-10-01790-f006:**
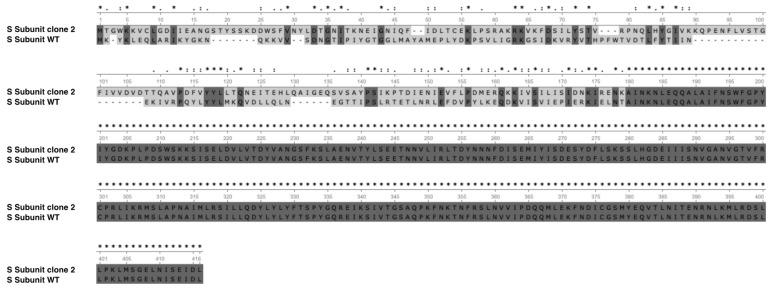
Protein sequence alignment of the apparent complete S subunit of RM type I system for WT and clone 2 strains. * indicates conserved aminoacids. Clones 1, 2 and 3 produce the same S subunit of RM type I system.

**Figure 7 microorganisms-10-01790-f007:**
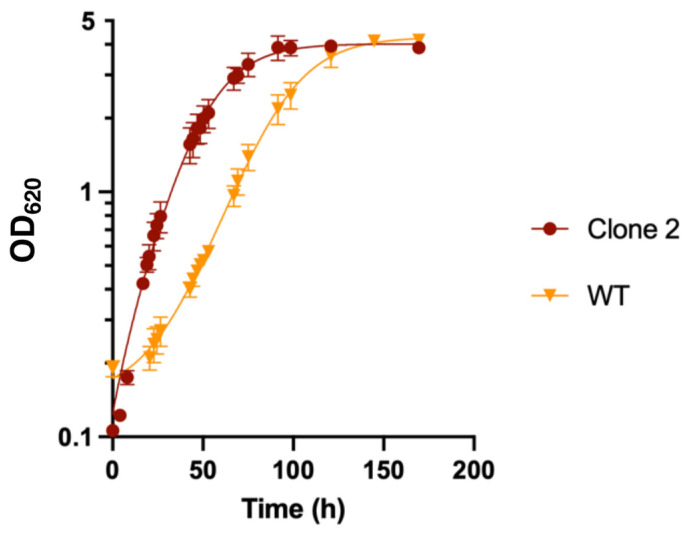
Growth profile of the WT and clone 2 strains in methanol synthetic medium supplemented with 0.5 g·L^−1^ YE. Cultures were performed in triplicate and error bars indicates standard deviation.

**Figure 8 microorganisms-10-01790-f008:**
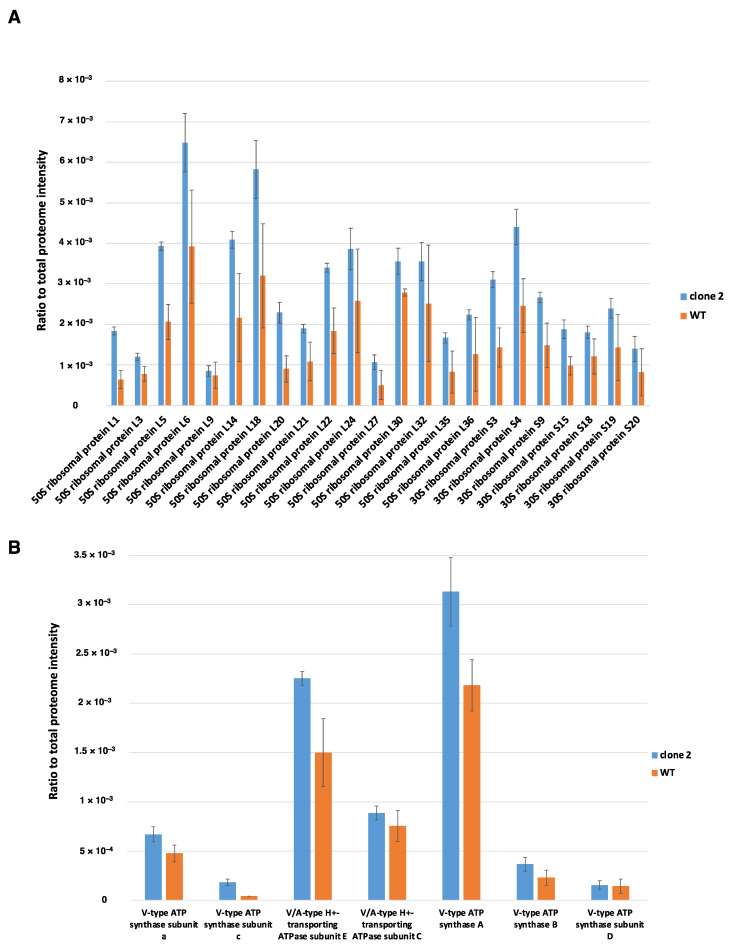
Assessment of the total proteome extraction. (**A**) Assessment of the total cytosolic protein extraction by comparison of all ribosomal proteins extracted. (**B**) Assessment of total proteome extraction on membrane bound proteins, example of ATPase complex. Data expressed in proportion of total proteome intensity. Blue bars correspond to clone 2 strain, and orange bars correspond to the WT strain.

**Figure 9 microorganisms-10-01790-f009:**
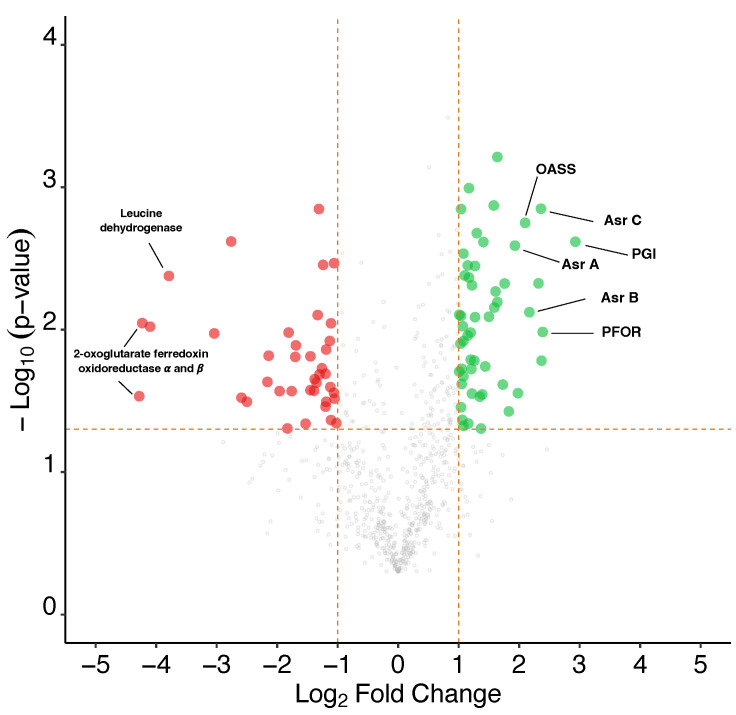
Volcano plot representing the results of total proteome analysis between the clone 2 strain against the WT strain. Green dots show significantly overproduced proteins, while red dots show significantly underproduced proteins. Horizontal dotted line represents the applied significance threshold of a one-sided student’s *t*-test *p* value of 0.05. Vertical dotted lines represent the applied threshold of a log_2_ fold change of >1 and <−1, corresponding to an absolute fold change >2 or <−2. Asr, anaerobic sulphite reductase; OASS, O-acetyl-homoserine sulphhydrylase; PFOR, pyruvate ferredoxin oxidoreductase; PGI, glucose-6-phosphate isomerase.

**Figure 10 microorganisms-10-01790-f010:**
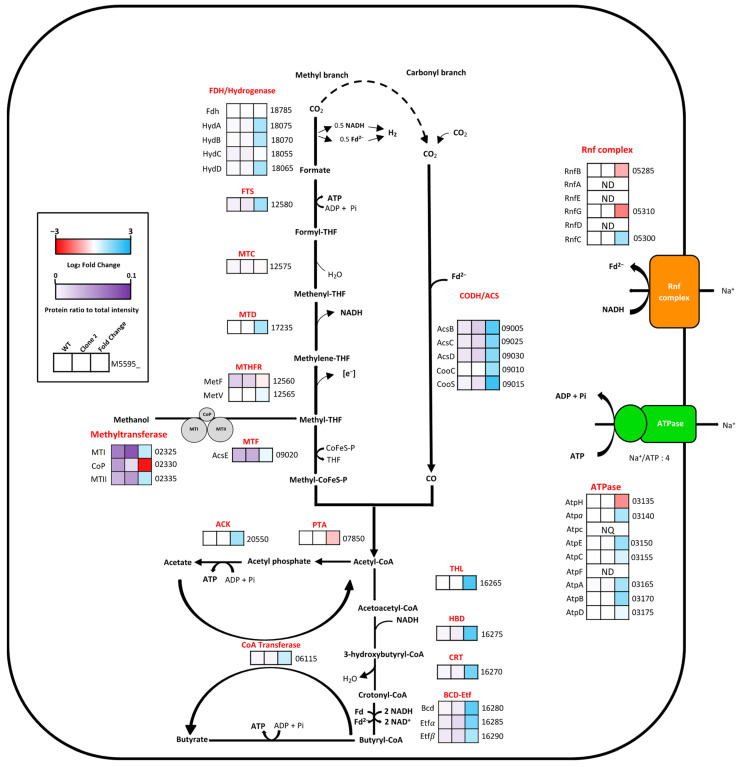
Comparative enzyme abundance in the central carbon metabolism between the WT and the adapted strain clone 2 in methanol synthetic medium. The first and second boxes correspond to the relative protein abundance expressed as a proportion of total intensity value of the WT and clone 2 strains, respectively. The third box corresponds to the Log_2_ fold change of protein abundance of the clone 2 strain relative to the WT strain. The number indicated at the right of each box corresponds to the *E. limosum* B2 locus tag, starting with M5595; [e^−^], undefined reducing equivalent; NQ, not quantified; ND, not detected. List of proteins is available in Appendix A.

**Figure 11 microorganisms-10-01790-f011:**
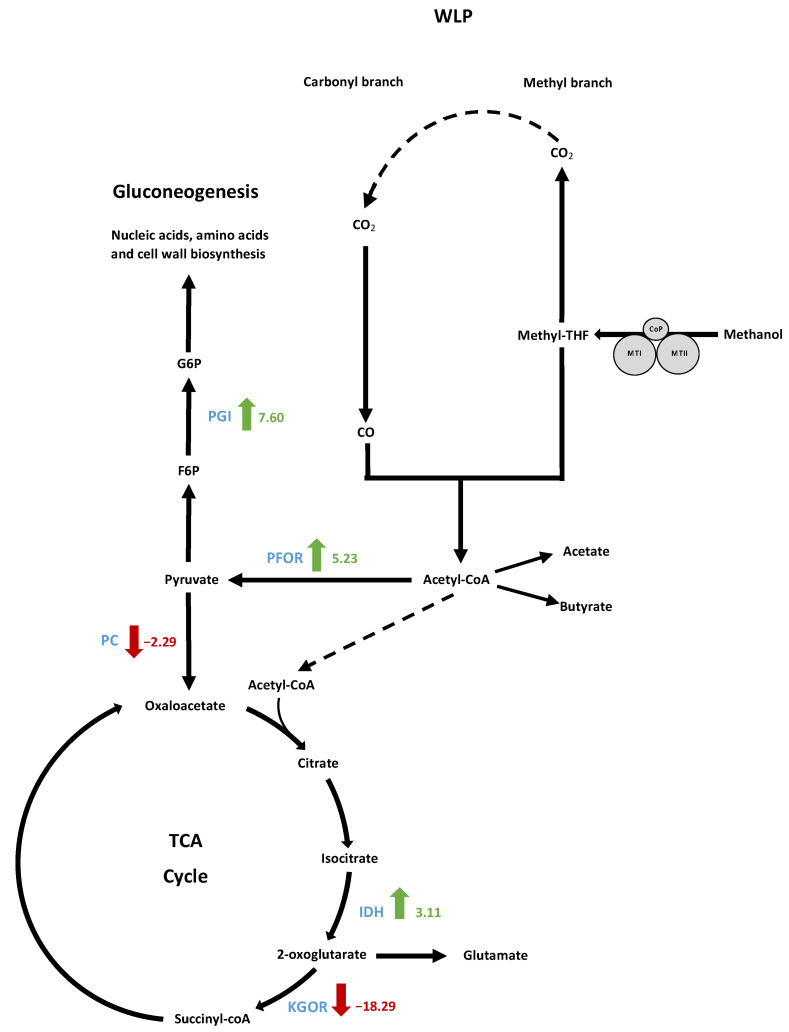
Schematic pathway showing significatively differentially produced proteins in clone 2 strain compared to WT strain in gluconeogenesis and tricarboxylic acid cycle alongside WLP. Pathways were simplified to show only proteins of interest. Green arrows indicate protein overproduction and red arrows indicate underproduction in clone 2 strain compared to WT strain. The scores indicate the fold change values. IDH, isocitrate dehydrogenase; KGOR, 2-oxoglutarate ferredoxin oxidoreductase; PC, pyruvate carboxylase; PFOR, pyruvate ferredoxin oxidoreductase; PGI, glucose-6-phosphate isomerase.

**Figure 12 microorganisms-10-01790-f012:**
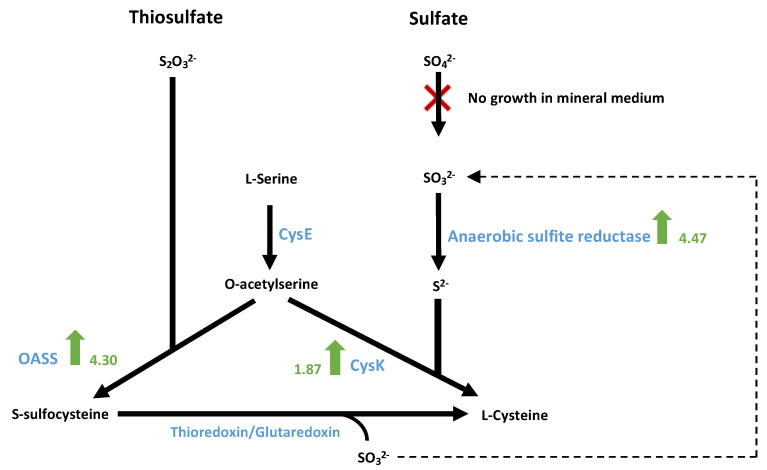
Theoretic pathway for sulphur assimilation and cysteine biosynthesis in *E. limosum* B2 in mineral medium. Green arrows indicate protein overproduction in clone 2 strain compared to WT. The scores indicate the fold change values. CysE, serine transacetylase; CysK, cysteine synthase; OASS, O-acetyl-homoserine.

**Table 1 microorganisms-10-01790-t001:** Genomic differences between the *Eubacterium limosum* B2 and ATCC 8486 strains.

Locus Tag ATCC8486	Locus Tag B2	Position B2	Position ATCC8486	Gene	Nucleotide Variation ATCC8486/B2	Type	AA Change	Description
B2M23_RS00035	M5595_17445	3667592	4431	-	G1265T	SNP	Ala^421^Glu	Phage terminase
B2M23_RS00620	M5595_16850	3547040	124983	-	A514G	SNP	Tyr^171^His	Acyl ACP thioesterase
B2M23_RS00770	M5595_16695	3512058	159965	-	C1065T	SNP	Val^355^Val	Transcriptional activator
Intergenic region	Intergenic region	2725372	946652	-	G/A	SNP	-	-
Intergenic region	Intergenic region	3264417	407607	-	A/-	DEL	-	-
B2M23_RS06095	M5595_11305	2338369	1333655	*cheY*	A187G	SNP	Arg^63^Gly	Response regulator
B2M23_RS08865	M5595_08530	1755671	1916353	*aspS*	G1004A	SNP	Gly^335^Glu	Aspartate--tRNA ligase
B2M23_RS08970	M5595_08425	1735036	1936988	-	G1007T	SNP	Thr^336^Asp	Bacitracin export permease
Intergenic region	Intergenic region	1442364	2231325	-	A/C	SNP	-	Lysin riboswitch
B2M23_RS11215	M5595_20825	4406797	2387123	-	-/G	INS	-	Hypothetical protein
Intergenic region	M5595_20830	4406887	2387213	-	-/A	INS	-	Hypothetical protein
Intergenic region	M5595_20830	4406916	2387242	-	-/T	INS	-	Hypothetical protein
B2M23_RS13215	M5595_19385	4089680	2815304	-	G234T	SNP	Met^78^Ile	Mobilization protein
Intergenic region	M5595_00360	73507	3183052	-	-/C	INS	-	Restriction endonuclease subunit S
Intergenic region	M5595_00360	73556	3183101	-	-/TG	INS	-	Restriction endonuclease subunit S
Intergenic region	M5595_00360	73990	3183516	-	-/G	INS	-	Restriction endonuclease subunit S
Intergenic region	Intergenic region	367145	3476696	-	T/-	DEL	-	-
B2M23_RS17820	M5595_03295	694242	3803795	*ktrA*	A/-	DEL		TrkA family potassium uptake protein
B2M23_RS18850	M5595_04320	887882	3997435	*pucK*	C/T	SNP	Ile^116^Thr	Uric acid permease PucK/Xanthine permease
B2M23_RS18850	M5595_04320	887885	3997438	*pucK*	A/G	SNP	Gly^117^Asp	Uric acid permease PucK/Xanthine permease
B2M23_RS19910	M5595_05415	1121402	4230955	*pbpF*	T/C	SNP	Ser^459^Pro	Transglycosylase domain-containing protein

**Table 2 microorganisms-10-01790-t002:** Summary of methylome data from the *E. limosum* B2 and the evolved clone 2 strains.

Strain	Motif	Predicted RM Type	Modified Position	Methylation Type	Fraction (%)	Number of Methylated Motifs Detected	Number of Total Motifs Detected
*E. limosum* B2	GCGRAG	II	5	m6A	67.6	2058	3042
	CAAAAAR	II	6	m6A	37.3	1623	4350
	CNNTAYNNNNNTCC	I	5	m6A	27.9	293	1052
*E. limosum* B2 clone 2	GCGRAG	II	5	m6A	81	2464	3042
	CAAAAAR	II	6	m6A	83.7	3642	4350
	CNNTAYNNNNGTG	I	5	m6A	79.8	576	1052

**Table 3 microorganisms-10-01790-t003:** List of mutations observed for the 10th, 25th, 50th, and 75th generations and Clones 1, 2 and 3 during ALE against the WT strain of *E. limosum* B2. Blue indicates a high rate of mutation among the considered population, while red indicates a low rate of mutation. The numbers represent the percentage of the population carrying genetic changes compared to the WT strain.

Locus Tag *E. limosum* B2	Position	WT	Variant	Type	10th	25th	50th	75th	C1	C2	C3	Gene	Gene in 5′ of Intergenic Region	Locus tag 5′ Gene Intergenic Region	Gene in 3′ of Intergenic Region	Locus tag 3′ Gene Intergenic Region
M5595_20825	4406858	-	T	INS	100	100	100	100	100	100	100	Hypothetical protein	-	-	-	-
M5595_20830	4406927	-	T	INS	100	100	100	100	100	100		Hypothetical protein	-	-	-	-
M5595_00020	4683	G	A	SNP	100		100	100	100		100	Hypothetical protein	-	-	-	-
M5595_04930	1017540	C	A	SNP	100					100	100	Hypothetical protein	-	-	-	-
M5595_17525	3682586	A	C	SNP	36.2		22					Hypothetical protein	-	-	-	-
-	1193634	A	-	DEL	58.9							Intergenic region	Ketol-acid reductoisomerase (3′ -> 5′)		Alpha-amylase family glycosyl hydrolase (3′ -> 5′)	
M5595_16535	3470810	G	A	SNP	100							Sugar binding transcriptionnal regulator	-	-	-	-
M5595_12615	2631029	T	C	SNP	100							HAD-IA family hydrolase	-	-	-	-
M5595_17525	3682598	C	T	SNP	31.8							Hypothetical protein	-	-	-	-
-	4262268	C	-	DEL	100							Intergenic region	Threonine synthase (3′ -> 5′)	M5595_20065	Homoserine dehydrogenase (5′ -> 3′)	M5595_20070
M5595_06115	1274807	T	C	SNP	100							Butyryl-CoA:acetate CoA-transferase	-	-	-	-
M5595_11550	2401587	C	T	SNP		75.5						Prepilin peptidase	-	-	-	-
M5595_08805	1810120	T	-	DEL			95.9					Hypothetical protein	-	-	-	-
-	2743930	A	-	DEL			43.4					Intergenic region	Helix-turn-helix transcriptional regulator (5′ -> 3′)	M5595_13160	Lysozyme family protein (5′ -> 3′)	M5595_13165
M5595_15025	3141664	AT	-	DEL			37					Arsenate reductase family protein	-	-	-	-
M5595_10115	2083561	A	-	DEL			62.1					Glycosyltransferase family 2 protein	-	-	-	-
-	3378354	T	C	SNP				70.2				Intergenic region	Rrf2 family transcriptional regulator (3′ -> 5′)	M5595_16095	FtsX-like permease family protein (5′ -> 3′)	M5595_16100
-	2164707	T	G	SNP				70.2				Intergenic region (riboswitch)	Leucine-rich repeat domain-containing protein (5′ -> 3′)	M5595_10385	Energy-coupling factor ABC transporter permease (3′ -> 5′	M5595_10390
M5595_09145	1871776	AACTG	-	DEL				25				Cysteine synthase A	-	-	-	-
-	4051265	T	-	DEL				88.2				Intergenic region	HPr family phosphocarrier protein (5′ -> 3′)	M5595_19190	Aminodeoxychorismate/anthranilate synthase component II (5′ -> 3′)	M5595_19195
M5595_03625	755658	C	A	SNP				35.2				Peptidoglycan DD-metalloendopeptidase	-	-	-	-
-	825607	T	-	DEL				46.7				Intergenic region	PucR family transcriptional regulator (5′ -> 3′)	M5595_03980	DUF917 domain-containing protein (5′ -> 3′)	M5595_03985
M5595_00050	9361	T	-	DEL				93.3				Hypothetical protein	-	-	-	-
-	370141	A	-	DEL				50	100	100		Intergenic region	ParB/RepB/Spo0J family partition protein (5′ -> 3′)	M5595_01730	Hypothetical protein (5′ -> 3′)	M5595_01735
M5595_05545	1145373	T	C	SNP						100		Nucleotide sugar dehydrogenase	-	-	-	-
M5595_10115	2082950	A	-	DEL							100	Glycosyltransferase family 2 protein	-	-	-	-
-	3881186	T	C	SNP				25				Intergenic region	tRNA-Phe (5′ -> 3′)	M5595_18430	Cobalamin-dependent protein (5′ -> 3′)	M5595_18435
M5595_00360	73656	-	T	INS							100	Restriction endonuclease subunit S	-	-	-	-

## Data Availability

Methylome, raw reads from whole-genome sequencing and raw proteomics data are available upon request. The annotated genome sequence of *E. limosum* B2 WT was deposited in Genbank (NCBI) under the accession number CP097376.1.

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
