# Peer review of "Genome Sequence of Eubacterium limosum B2 and Evolution for Growth on a Mineral Medium with Methanol and CO2 as Sole Carbon Sources"

_microorganisms, 2022, doi:10.3390/microorganisms10091790_

Round 1

Reviewer 1 Report

Eubacterium limosum is a fascinating acetogenic bacterium that can produce butyrate and acetate via methanol fermentation. The authors experimentally adapted E. limosum B2 to defined medium and sequenced genomes at intervals.  The manuscript is well-written and the study is overall very well-done. The lack of a mechanistic explanation for the growth phenotype of adapted isolates is unsatisfying, as it is still open-ended if epigenetics or a chromosomal mutation(s) are responsible for the enhanced growth that was observed. The following are questions and suggestions to improve the paper.

1.       What is the mutation rate? How many mutations were expected by 100 generations? Was the population bottleneck sufficient by 100 generations to select for fixed mutations?

2.       Statistics should be clarified in every figure legend: error bars represent what? Number of biological and technical replicates, etc.

3.       L 120. What was the ratio of inoculum to culture volume?

4.       L 253. Is this of just 16S gene?

5.       Fig 1 colors are too difficult to distinguish, text size is too small to read even when magnified as a pdf. I am not sure how useful this tree is? Perhaps you can prune/collapse the leaves and root with an outlier?

6.       Fig 2 blurry. The genetic orientation of the rearrangements is not clear. Where is the major ori?

7.       L 332-334. Some nutrients are only necessary in pmol-nmol concentration. It is doubtful two subcultures eliminated carryover. Please revise this sentence accordingly.

8.       Fig 3 blurry. The y axis is labeled incorrectly. It should read “Optical Density (620nm)” or similar. We do not measure absorbance using this technique, we measure optical density, and it is not the same thing.

9.       Fig 4 blurry. What are the units on panel B y axis? Please define. The abbreviation for methanol is MeOH. Please change throughout.

10.   Fig 5. See comment above about y axis labels.

11.   Figures 4b, 5b: did I read that correctly that only duplicate technical replicates were used to quantify metabolites? There should be minimum triplicate biological replicates in order for the statistics to be meaningful. Please provide number of technical and biological replicates in the figure legends throughout.

12.   Fig 6 text is too small.

13.   Fig 7 blurry. Text is too small.

14.   Fig 8 blurry. See comment above about y axis labels.

15.   L 544. Superscript “YE” should be full-sized.

16.   L 546. I question the choice of using ribosomal proteins as a calibrant. In most organisms, faster growth is associated with higher ribosome content. How do you control for that? And as it seems metabolic efficicency is increased, ATP synthase might not be a good calibrant for membrane proteins, either. What happens if you compare expression levels using random cytoplasmic/membrane proteins? The discussion later in the paragraph proves my point! Please revise this section to clarify that you really looked at global average abundance.

17.   Fig 9 blurry. Panel a is unreadable. 

18.   L 595. Was this a Student’s T-test used for the p value? Instead, should use false-discovery rate…

19.   Fig 10 has a lot of white space. Suggest making dots larger, adjusting axes, and labeling some of the most interesting hits.

20.   Fig 11 is very nice, but some of the text is too small to read.

21.   L 642. You did not measure C flux through metabolism. Please revise the text throughout. One must remember that transcriptomics shows changes in transcript abundance, but often does not reflect proteome changes, and proteome changes do not necessarily indicate metabolic flux changes! In point of fact, as your fermentation product ratios were not significantly different from the parent.

22.   It would be helpful to the reader to create figures explaining the gluconeogenesis, sulfur metabolism, and other reactions that were differentially expressed. How do these relate to WLP?

23.   You had three clones that grew better than wild type. Clone 2 is similar to clone 1 and doesn’t seem interesting on the nucleotide level. Clone 3 has the interesting restriction enzyme difference, which is the only thing that looks like it might have a mechanistic basis for the phenotype. Why did you use clone 2 for proteomics?

24.   Genetic tools are available for E. limosum ATCC8486 (https://www.sciencedirect.com/science/article/pii/S1096717622000520), KIST612 (https://www.sciencedirect.com/science/article/pii/S2589014X20300736), and in https://pubs.acs.org/doi/10.1021/acssynbio.9b00150. Have you attempted to try these techniques in B2? If feasible, testing the hypotheses you generate would greatly enhance the impact of the manuscript. For instance, reconstructing any of these mutations in the parental background, or mutating DNA methylation sites to test the epigenetic hypotheses.

25.   Did you try reverting the adaptations by passaging back to rich medium? This experiment or complementation is necessary to determine if epigenetics or chromosomal mutations are responsible for the observed phenotypes.

Reviewer 2 Report

This is an excellent piece of research primarily around adaptive laboratory evolution in a butyrate-producing acetogen.  There is significant supporting data to explain observed effects.  This is the first time that I have seen an epigenetics analysis done in an acetogen.

I am providing some minor suggestions to improve the ms..

line 33  "(C1) substrates, ....".

line 34  "such as synthesis gas....".

line 56  "performed in the....".

line 57  "which does not produce....".

line 95  "serum bottles".

line 99  The authors my want to use free cys rather than the HCl salt for future work.  Free cys was not readily available 20 years ago but it is now.

lines 104-106  Better to just reference the trace metal solution used.  The boric acid, required by algae/plants, may be removed.  What about tungstate?

line 138  Remove the Oxford comma after "mobile phase".

line 151  The comma after "extraction yield" may also be omitted.

Figure 1  Please provide a distance bar for the tree.

lines 417-419  It's not a good idea to incubate cultures/plates in an anaerobic chamber.  There are a number of ways to do this outside of the chamber including wide moth jars with a 20 mm stopper fitted in the plastic lid, bear bags (I use OPSAK) which are impermeable to air (and odors which could attract bears) etc.  You can control the atmosphere in the containers by flushing and/or injecting known quantities of pure gases.

line 554  "a larger amount of....".

The References need work and the authors must take responsibility for this.

Some refs are missing page numbers, e.g. refs 6 and 9.

Ref 10  "pH".

Refs 19, 41 and 55 need subscripts.

Refs 29, 36 and 43 need italics and correct capitalization.

Round 2

Reviewer 1 Report

Thank you for adjusting figures and including statistical details in the figure legends. Unfortunately it seems as if Figure 9 did not load correctly to the pdf. In addition, the authors do not include complementation or mutagenesis studies to test any hypotheses for growth rate enhancement of the strain; they do not provide data from the appropriate passaging controls experiments to correlate the observed phenotype with the fixed genetic mutations, and no evidence for epigenetic modifications is shown. These data are fundamental to testing their proposed hypothesis. As it stands, the conclusions are correlative and the authors do not provide evidence for a mechanistic basis for their observations. In my opinion the work is premature for publication.
